# Cyclometallated Palladium(II) Complexes: An Approach to the First Dinuclear Bis(iminophosphorane)phosphane-[C,N,S] Metallacycle

**DOI:** 10.3390/molecules27207043

**Published:** 2022-10-19

**Authors:** Marcos Rúa-Sueiro, Paula Munín-Cruz, Adolfo Fernández, Juan M. Ortigueira, María Teresa Pereira, José M. Vila

**Affiliations:** Departamento de Química Inorgánica, Universidade de Santiago de Compostela, E-15782 Santiago de Compostela, Spain

**Keywords:** cyclometallation, palladium, iminophosphoranes, NMR, crystal structure

## Abstract

Treatment of bis(iminophosphorane)phosphane ligands **2a**–**2e** with Li_2_PdCl_4_ gave a set of novel diphosphane-derived complexes bearing two metallacycle rings, each one enclosing a P=N double bond: the unprecedented bis(iminophosphorane)phosphane-[C,N,S] palladacycles. In the case of the ligand derived from bis(diphenylphosphino)methane, **2a**, both the single and the double palladacycle complexes were obtained. Reaction of **3a** with bis(diphenylphosphino)ethane did not yield the expected product with the diphosphane bonded to both palladium atoms, but rather the novel coordination compound **5**. The crystal structures of **3c** and **5** are described.

## 1. Introduction

The chemistry of cyclometallated compounds, with particular emphasis on the palladacycles, has been of great relevance since their beginning, mostly due to the applications they present: specifically, they behave as pre-catalysts in cross-coupling reactions with C-C bond formation, as, for example, the Suzuki–Miyaura [1,2] or the Mizoroki–Heck reactions [3,4]; they can present activity as antitumor or anticancer agents [5,6,7], with some displaying activity similar to cis-platinum. They may also have interesting luminescent properties [8,9]. We have extensively studied the synthesis and characterization of palladacycles [10] and more recently those containing thiosemicarbazone [11,12,13] and Schiff base ligands [14,15], giving further insights into their chemistry. On the other hand, iminophosphoranes, also known as phosphazenes, are a type of organic ligand of general formula R_3_P=NR’ widely used in coordination chemistry [16,17,18,19]. They may incorporate additional donor atoms becoming quite versatile ligands; they are synthesized by the Staudinger reaction [20]. In recent years, iminophosphorane chemistry has undergone a remarkable breakthrough [21] inclusive of the functionalized iminophosphoranes useful in homogeneous catalysis, which may be conveniently modified to achieve the most desirable properties of the corresponding complexes [22,23]. These ligands are a very good choice for synthesizing cyclometallated compounds, and have been known for quite some time [24]. The main feature that makes them optimal for coordinating to metal centers is the polarity in the P=N double bond. Nitrogen bears a negative partial charge inducing a strong σ donor and weak π acceptor character which allows the formation of strong bonds with palladium. Moreover, these ligands are known to also coordinate to metal centers through the carbon in the ortho position to phosphorus to give preferentially endo species [25]. These ligands display other interesting properties: they can be used as synthetic intermediates in the Aza–Wittig reaction [26,27], generating heterocycles that act as intermediates in drug synthesis; they are also widely used as superbases [28] with a low nucleophilicity, which can be modulated according to their substituents. Additionally, they exhibit very low toxicity, and it has been found that when coordinated to transition metals they can exhibit high antitumor activity [29,30].

In Figure 1, some relevant examples of iminophosphorane palladacycles are depicted which show the terdentate [C,N,S] aniline complex by Grévy [31] I, the [C,N,S] [21] and [C,N] [16] structures II and III by Urriolabeitia, and the terdentate [C,N,O] [32] IV and tetradentate [C,N:C,N] [33] V dinuclear complexes by Vila. In all cases, the corresponding iminophosphorane was prepared from the azide by using a PR_3_ ligand, where appropriate, in the ensuing palladacycles, with a bridging diphosphine; the latter holds together the two palladium iminophosphorane moieties. Likewise, although the related bis(iminophosphorane) ligands are known [33,34,35,36], the ensuing dipalladacycles VI are to the best of our knowledge still outstanding. Hence, at variance with the hitherto known examples, the preparation of the first bis(iminophosphorane)phosphane palladacycle complexes seemed to be a challenging quest as well as a new approach to this chemistry for a number of reasons: (a) in type VI complexes, the phosphorus atoms are multifunctional, being part of the palladacycle ring and of the bridging diphosphane; (b) their number is brought down to two, cf. to the previous known examples, where there were two sets of independent phosphorus nuclei. [32]; (c) additional reactivity can be carried out, e.g., by exchanging the terminal chloride ligands for mono- or diphosphanes, the latter case opening the possibility of folding the starting dinuclear structure of the complex. The results of such a process are reported herein.

## 2. Results and Discussion

For the convenience of the reader, the compounds and reactions are shown in Figure 2. The compounds described in this paper were characterized by elemental analysis (C, H, N, S); IR, ^1^H and ^31^P−{^1^H} NMR spectroscopy (see the Section 4); and by X-ray single crystal diffraction. Ligands **2a**–**2e** were readily synthesized by the Staudinger reaction from the corresponding triphenylphosphane and 2-thiomethyl azide. The sp^2^ carbon atoms available for metallation are on the phosphorus phenyl rings, giving the *endo* type species. The IR spectra show the υ(P=N) stretch ca. 1330 cm^−1^ and total absence of any band assignable to azide vibrations modes. The ^1^H and ^31^P−{^1^H} NMR resonances were assigned accordingly (see Section 4), the latter showing a singlet resonance at 0–10 ppm in agreement with equivalent phosphorus nuclei. Treatment of **2a**–**2e** with Li_2_PdCl_4_ gave the hitherto unknown bis(iminophosphorane)phosphane-[C,N,S] palladacycles **3a**–**3e**.

Thus, treatment of **2a** with Li_2_PdCl_4_, made in situ from LiCl and PdCl_2_, in a 2:1 molar ratio and sodium acetate in methanol gave an untreatable mixture, which was not further pursued. However, analogous treatment of **2a** in a 1:2 molar ratio in refluxing methanol for 18 h gave a solution and a solid which were conveniently separated. From the solution, the resulting residue after solvent removal was recrystallized from dichloromethane to give the single palladacycle complex **4**; the solid was dissolved in dichloromethane, filtered through silica gel and after work up recrystallized from hexane to give the double palladacycle complex **3a**. Further attempts to prepare **3a** or **4** independently were unsuccessful. Reaction of **2b**–**2e** with Li_2_PdCl_4_ in a 1:2 molar ratio and sodium acetate in methanol with stirring gave complexes **3b**–**3e** as yellow to pale-yellow air-stable solids; single palladacycle complexes were not observed in any of the reactions. We assume that the differing behavior of ligand **2a** in **3a** as opposed to **2b**–**2c**, in **3b**–**3c** could be due to certain steric hindrance imposed by the shorter PC_n_P carbon chain in the former case, making *stronger* reaction conditions necessary to produce double metalation, i.e., long refluxing hours, whereas in the latter cases, a milder process at r.t. readily gives the double-nuclear palladacycle. For **3d** the PCP carbon is a sp^2^ carbon, hence with a greater PCP bond angle of 120°, cf. 109.5° **3a**, thus lowering the steric barrier. For **3e**, in spite of the theoretical PNP angle of 109.5° for the N(sp^3^), only coordination of the diphosphine in the bridging mode was observed. Although presently we have no sound explanation for this behavior, this seems to be a rather frequent arrangement of the phosphine (see the SI section for references regarding this issue). The single palladated complexes have so far not been pursued further in the hope of finding a suitable synthetic procedure to give the pure compounds. Notwithstanding, we have observed a rather mild shift in the non-metalated P=N group at 41.70 ppm: the possibility of a P=O group in view of the microanalytical and spectroscopic data, which are in accordance with the proposed formulation for compound **4**.

The IR spectra show the υ(P=N) band ca. 1220 cm^−1^ shifted by ca. −100 cm^−1^ compared to the starting ligand, as well as the band assigned to the υ(Pd–Cl) vibration mode ca. 325 cm^−1^, in accordance with a terminal chloride ligand. In the ^1^H NMR spectra resonances, ca. 4.5 (doublet of doublets) and 2.63 (broad singlet) ppm were assigned to the PCH_2_P **3a** and PN(Me)P **3e** protons, respectively, whereas the P(CH_2_)_2_P **3b** and P(CH_2_)_3_P **3c** resonances were multiplets. In the case of **3d**, the C(=CH_2_) resonances were two apparent triplets for the proton part of the AA’XX’spin system generated by the P_2_C(=CH_2_) nuclei. As for the complex **4**, the PCH_2_P resonance (proton ABXY system) was an apparent doublet at 3.26 ppm. The H5 resonance was high field shifted with respect to the parent ligand and showed coupling to the phosphorus nuclei (^4^JPH5 ca. 10 Hz). The ^31^P NMR spectra showed a singlet resonance for the two equivalent phosphorus nuclei ca. 48–58 ppm. For complex **3c**, for which two singlets ca. 57 ppm were observed, we suggest the presence of two isomers syn and anti, Figure 3. Likewise, the ^1^H NMR spectrum for **3c** also showed two similar sets of proton resonances. For complex **4**, there were two singlet resonances at 41.70 and 51.36 ppm in the ^31^P NMR spectrum, consequent of two non-equivalent phosphorus nuclei.

Suitable crystals of **3c** were grown by slowly evaporating an acetone solution of the compound. Crystal data are given in the Supporting Information. The ORTEP illustration of complex **3c** is shown in Figure 1. The compound crystalizes in the orthorhombic space group Pnma with two acetone solvent molecules. The crystals consist of discrete molecules separated by van der Waals distances, bearing two palladacycle moieties symmetry related by a 2-fold axis passing through the C(21) carbon atom of the iminophosphorane ligand. The crystallographic independent palladium atom is bonded to four different atoms in a slightly distorted square—planar disposition: a P=N nitrogen atom, an aryl carbon atom, a sulfur from the thiomethyl aniline, and finally by a chloride ligand. The metal coordination planes [C,N,S,Cl] form a dihedral angle of 22.24°. All bond lengths and angles are within the expected values, with allowance for lengthening of the Pd(1)–S(1) bond, 2.358 Å. The bond angles at palladium are close to 90° with the smaller values for the bite angles for the tridentate ligand [C(1)-Pd(1)-N(1) = 88.5° and N(1)-Pd(1)-S(1) = 84.99°] consequent upon chelation and the sum of angles at palladium is 360.16°. The large distance between the metallated rings and the thioaniline rings precludes any π-π stacking interactions between the metalated rings.

In view of the arrangement of the rings at palladium in the crystal structure of **3c**, our ensuing hypothesis was that it could be possible to link the metal atoms of the folded compound with a diphosphane after removing the chloride ligands. For this purpose, compound **3a** was chosen, having the shortest carbon chain between the phosphorus atoms; also, considering the deviation of the metallated rings from their ideal parallel arrangement (probably more increased in solution), Ph_2_P(CH_2_)_2_PPh_2_ (dppe) was selected as the attempted binding diphosphane between the metal centers. The resulting species would be related to the known A-frame structures [14]. Thus, reaction of complex **3a** with dppe and ammonium hexafluorophosphate in acetone under stirring condition for 18 h at room temperature gave a sticky solid, for which the spectroscopic data indicated to be a complex mixture of products, which were not further pursued. However, the chloroform solution of this solid gave suitable crystals for X-ray structural analysis that proved to be not the expected product with the diphosphane bonded to both palladium atoms, but rather a different and new coordination compound, labelled **5**. (see Figure 2).

Compound **5** crystallizes in the P2_1_/n space group with two chloroform solvent molecules; the ORTEP illustration is shown in Figure 2. The molecular structure of **5** consists of discrete molecules with the palladium atom bonded to two pairs of nitrogen and sulfur atoms in a cis arrangement, pertaining to a tetradentate-[N,S,N,S] bis(iminophosphorane) ligand, in a distorted square-planar geometry. The sum of coordination bond angles at palladium is 359.67°. As for the bond distance, the most noteworthy feature is the somewhat longer Pd-N distance of 2.075 Å compared to the analogous bond length in **3c,** consequent on the greater trans influence of the sulfur atom in **5** as opposed to the chloride ligand in **3c**. Similarly, the Pd(1)-S(1) bond length, 2.267 Å, is slightly shorter due to the stronger trans influence of the phenyl carbon atom in **3c** vs. the P=N nitrogen atom in **5**. The PCP and palladium coordination planes, [P(1)C(15)P(2)] and [N(1)N(2)S(1)S(2)], respectively are at an angle of 74.43°; whilst the two rings at palladium [Pd(1)S(1)C(6)C(5)N(1)] and [Pd(1)S(2)C(13)C(12)N(2)] are at 22.8°. In relation to the similarity of the structures, it should be noted that there are two phosphorus atoms pertaining to a single six-membered coordination ring and four non-substituted phenyl rings in **5**, as opposed to two phosphorus atoms, each one embedded in a metallacycle ring and two non-substituted phenyls in **3c**, highlighting the difference in the diphosphane as a bidentate ligand vs. its bis(iminophosphorane) counterpart. Even assuming that complex **5** has been obtained serendipitously, representing an intriguing reactivity, and that it seems to stem from a re-protonation of the palladium-phenyl moiety of complex **3a,** two issues may be considered. First of all, if complex **5** can be prepared directly from the reaction of the iminophosphorane precursor **2a** and the palladium(II) reagent without the added base; secondly, to provide a reasonable answer, or at least a tentative route, of the formation for **5**. As for the former question, this would provide a satisfactory explanation and solve the intrigue in its formation, however leading to a much less interesting chemistry. Nevertheless, all attempts to the mentioned synthesis of complex **5** via this route have been to no avail, which brings us to the second point, in other words, of finding or proposing a plausible explanation for this new reactivity pattern; understanding that the process should be made repeatable to determine the proper reaction conditions. This poses quite a challenge, as with many by chance reactions. Notwithstanding, and in the absence of further data at this time, we suggest that after chloride abstraction by ammonium hexafluorphosphate and re-protonation of the phenyl ring, one Pd(II) is chelated by the dppe phosphane; ligand **2a** then coordinates to the second Pd(II) center to give complex **5**. This research, together with the reactivity of complexes **3b**–**3e** with dppe and related diphosphanes, is currently underway.

## 3. Conclusions

In conclusion, a new family of bis(iminophosphorane) metallacycles derived from diphosphanes, which contain two five-membered metallacycle rings in the endo arrangement, have been successfully synthesized. The phosphorus atoms are multifunctional, being part of the palladacycle ring and of the bridging diphosphane For this purpose, the synthetic method of cyclometallation using bis(iminophosphorane) ligands has been followed; the corresponding ligands were synthesized using the Staudinger reaction. The crystal structural analysis for **3c** confirms the spectroscopic results, providing satisfactory evidence regarding the orientation of the metallated rings, which show a close-to-parallel arrangement. Treatment of compound **3a** with the ammonium hexafluorophosphate salt and diphosphine dppe did not give the hoped dinuclear complex with bridging dppe, but rather the formation of the chelate coordination compound **5**. The attempt to prepare complex **5** by direct treatment of **2a** and an appropriate palladium salt was unsuccessful at reproducing the result here reported. Further studies are presently in progress in order to shed more light on this process and those concerning complexes **3b**–**3e**.

## 4. Experimental Section

### 4.1. General Procedures

Solvents were purified by standard methods [37]. The reactions were carried out under dry nitrogen. 2-(methylsulfanyl)aniline, Ph_2_PCH_2_PPh_2_ (dppm), Ph_2_P(CH_2_)_2_PPh_2_ (dppe), Ph_2_P(CH_2_)_3_PPh_2_ (dppp), Ph_2_PC(CH_2_)PPh_2_ (vdpp), lithium chloride and palladium chloride (II) were purchased from commercial sources; Ph_2_PN(Me)PPh_2_ (dppma) was a personal loan to Professor JM Vila. Elemental analyses were performed with a Thermo Finnigan elemental analysis and model Flash 1112. IR spectra were recorded on Jasco model FT/IR-4600 spectrophotometer. ^1^H NMR and spectra in solution were recorded in acetone-d_6_ or CDCl_3_ at room temperature on Varian Inova 400 spectrometers operating at 400 MHz using 5 mm o.d. tubes; chemical shifts, in ppm, are reported downfield relative to TMS using the solvent signal as reference acetone-d_6_ δ ^1^H: 2.05 ppm, CDCl_3_ δ ^1^H: 7.26 ppm). ^31^P NMR spectra in solution were recorded in acetone-d_6_ or CDCl_3_ at room temperature on Varian Inova 400 spectrometer operating at 162 MHz using 5 mm o.d. tubes and are reported in ppm relative to external H_3_PO_4_ (85%). Coupling constants are reported in Hz. All chemical shifts are reported downfield from standards.

### 4.2. Preparation of the Ligands


**Synthesis of 1**


2-(methylsulfanyl)aniline (500 mg, 3.59 mmol) was added in a mixture of water (2 cm^3^), ethyl acetate (8 cm^3^) and hydrochloric acid (1 cm^3^). The solution was cool below 5 °C and a cold solution of NaNO_2_ (421 mg, 6.10 mmol, 70% excess) was slowly added to the mixture, which was stirred for 30 min. Then, a cold solution of NaN_3_ (397 mg, 6.10 mmol, 70% excess) was slowly added, taking care to not overheat it above 5 °C, and the solution was stirred for 30 min. The reaction mixture was extracted with 3 × 10 cm^3^ of ethyl acetate and the organic phase was washed with water and dried under vacuum. Yield: 445 mg, 75%. Anal. calculated: C: 50.9, H: 4.3, N: 25.4, S: 19.4%; found: C: 50.7, H: 4.4, N: 25.3, S: 19.2%; C_7_H_7_N_3_S (165.21 g/mol); IR (cm^−1^): υ_as_(N_3_) 2123, υ_s_(N_3_) 1280. ^1^H NMR (acetone-d_6_, δ/ppm): 7.14–7.25 (m, 4H, H^1^H^2^H^3^H^4^), 2.42 (s, 3H, SCH_3_).


**Synthesis of 1,1′-methylenebis(N-(2-(methylthio)phenyl)-1,1-diphenyl-λ^5^-phosphanimine) 2a.**


A 2:1 mixture of MeSC_6_H_4_N_3_ (300 mg, 1.82 mmol) and Ph_2_PCH_2_PPh_2_ (349 mg, 0.91 mmol) in diethyl ether (15 cm^3^) was refluxed for 24 h, with N_2_ bubbling. After stirring for 24 h at room temperature, a yellow solid was formed, which was filtered off and dried under vacuum. Yield: 323 mg, 54%. Anal. calculated: C: 71.1, H: 5.5, N: 4.3, S: 9.7%; found: C: 71.1, H: 5.6, N: 4.3, S: 9.7%; C_39_H_36_N_2_P_2_S_2_ (658.80 g/mol); IR (cm^−1^): υ(P=N) 1332. ^1^H NMR (CDCl_3_, δ/ppm, J/Hz): 7.82 (dd, 8H, H^o^, ^3^JH^o^/P = 11.7, ^3^JH^o^/H^m^ = 7.7), 7.30 (m, 12H, H^m^/H^p^), 7.04 (d, 2H, H^4^, ^3^JH^4^/H^3^ = 7.2), 6.72 (m, 4H, H^2^/H^3^), 6.21 (d, 2H, H^1^, ^3^JH^1^/H^2^ = 7.2), 3.85 (m, 2H, PCH_2_), 2.48 (s, 6H, SCH_3_). ^31^P-{^1^H} NMR (CDCl_3_, δ/ppm): 22.37 (s, PPh_2_).


**Synthesis of 1,1′-(ethane-1,2-diyl)bis(N-(2-(methylthio)phenyl)-1,1-diphenyl-λ^5^-phosphan-imine) 2b.**


A 2:1 mixture of MeSC_6_H_4_N_3_ (100 mg, 0.61 mmol) and Ph_2_P(CH_2_)_2_PPh_2_ (121 mg, 0.30 mmol) in diethyl ether (10 cm^3^) was refluxed for 24 h, with N_2_ bubbling. After stirring for 24 h at room temperature, a white solid was formed, that was filtered off and dried under vacuum. Yield: 168 mg, 82%. Anal. calculated: C: 71.4, H: 5.7, N: 4.2, S: 9.5%; found: C: 71.3, H: 5.5, N: 4.1, S: 9.2%; C_40_H_38_N_2_P_2_S_2_ (672.83 g/mol); IR (cm^−1^): υ(P=N) 1325. ^1^H NMR (acetone-d_6_, δ/ppm, J/Hz): 7.76 (m, 8H, H^o^), 7.51 (m, 4H, H^p^), 7.44 (m, 8H, H^m^), 6.98 (d, 2H, H^4^, ^3^JH^4^/H^3^ = 6.6), 6.66 (dd, 2H, H^2^, ^3^JH^2^/H^1^ = 7.2, ^3^JH^2^/H^3^ = 6.9), 6.58 (dd, 2H, H^3^, ^3^JH^3^/H^2^ = 6.9, ^3^JH^3^/H^4^ = 6.6), 6.15 (d, 2H, H^1^, ^3^JH^1^/H^2^ = 7.3), 2.74 (d, 4H, PCH_2_, ^3^JH/P = 13.3), 2.37 (s, 6H, SCH_3_). ^31^P-{^1^H} NMR (acetone-d_6_, δ/ppm): 5.72 (s, PPh_2_).


**Synthesis of 1,1′-(propane-1,3-diyl)bis(N-(2-(methylthio)phenyl)-1,1-diphenyl-λ^5^-phosphan-imine) 2c.**


A 2:1 mixture of MeSC_6_H_4_N_3_ (100 mg, 0.61 mmol) and Ph_2_P(CH_2_)_3_PPh_2_ (125 mg, 0.30 mmol) in diethyl ether (10 cm^3^) was refluxed for 24 h, with N_2_ bubbling. After stirring for 24 h at room temperature, a yellow solid was formed, that was filtered off and dried under vacuum. Yield: 141 mg, 68%. Anal. calculated: C: 71.7, H: 5.9, N: 4.1, S: 9.3%, found: C: 71.2, H: 5.7, N: 4.3, S: 9.1%; C_41_H_40_N_2_P_2_S_2_ (686.85 g/mol); IR (cm^−1^): υ(P=N) 1331. ^1^H NMR (acetone-d_6_, δ/ppm, J/Hz): 7.68 (dd, 8H, H^o^, ^3^JH^o^/P = 12.0, ^3^JH^o^/H^m^ = 6.8), 7.33 (t, 4H, H^p^, ^3^H^p^/H^m^ = 6.6), 7.27 (m, 8H, H^m^, *N* = 6.7), 6.86 (d, 2H, H^4^, ^3^JH^4^/H^3^ = 6.7), 6.53 (m, 4H, H^2^H^3^), 6.15 (d, 2H, H^1^, ^3^JH^1^/H^2^ = 7.1), 2.71 (dd, 4H, PC*H_2_*, ^3^JH/P = 16.3, ^3^JH/H = 7.6), 2.18 (s, 6H, SCH_3_), 1.48 (m, 2H, PCH_2_C*H_2_*). ^31^P-{^1^H} NMR (acetone-d_6_, δ/ppm): 5.39 (s, PPh_2_).


**Synthesis of 1,1′-(ethene-1,1-diyl)bis(N-(2-(methylthio)phenyl)-1,1-diphenyl-λ^5^-phosphan-imine) 2d.**


A 2:1 mixture of MeSC_6_H_4_N_3_ (100 mg, 0.61 mmol) and Ph_2_PC(=CH_2_)PPh_2_ (121 mg, 0.31 mmol) in diethyl ether (10 cm^3^) was refluxed for 24 h, with N_2_ bubbling. After stirring for 24 h at room temperature, a yellow solid was formed, that was filtered off and dried under vacuum. Yield: 193 mg, 95%. Anal. calculated: C: 71.6, H: 5.4, N: 4.2, S: 9.6%, found: C: 71.3, H: 5.5, N: 4.2, S: 9.2%; C_40_H_36_N_2_P_2_S_2_ (670.81 g/mol); IR (cm^−1^): υ(P=N) 1339. ^1^H NMR (acetone-d_6_, δ/ppm, J/Hz): 7.69 (dd, 8H, H^o^, ^3^JH^o^/P = 12.2, ^3^JH^o^/H^m^ = 7.3), 7.55 (t, 4H, H^p^, ^3^JH^p^/H^m^ = 7.4), 7.46 (dt, 8H, H^m^, ^3^JH^m^/H^o^ = 7.3, ^3^JH^m^/H^p^ = 7.4, ^4^JH^m^/P = 2.9), 6.95 (d, 2H, H^4^, ^3^JH^4^/H^3^ = 6.7), 6.58 (m, 4H, H^2^H^3^), 6.15 (d, 2H, H^1^, ^3^JH^1^/H^2^ = 7.7), 4.55 (d, 1H, PCCH_2_, ^2^JH/H = 1.5), 4.18 (d, 1H, PCCH_2_, ^2^JH/H = 1.5), 2.38 (s, 6H, SCH_3_). ^31^P-{^1^H} NMR (acetone-d_6_, δ/ppm): 3.57 (s, PPh_2_).


**Synthesis of N-methyl-N’-(2-(methylthio)phenyl)-N-(N-(2-(methylthio)phenyl)-P,P-diphenyl-phosphorimidoyl)-P,P-diphenylphosphinimidic amide 2e.**


A 2:1 mixture of MeSC_6_H_4_N_3_ (50 mg, 0.30 mmol) and Ph_2_PNCH_3_PPh_2_ (61 mg, 0.15 mmol) in diethyl ether (10 cm^3^) was refluxed for 24 h, with N_2_ bubbling. After stirring for 24 h at room temperature, a red solid was formed, that was filtered off and dried under vacuum. Yield: 65.7 mg, 64%. Anal. calculated: C: 69.5, H: 5.5, N: 6.2, S: 9.5%, found: C: 69.9, H: 5.7, N: 6.1, S: 9.2%; C_39_H_37_N_3_P_2_S_2_ (673.81 g/mol); IR (cm^−1^): υ(P=N) 1342. ^1^H NMR (acetone-d_6_, δ/ppm, J/Hz): 7.95 (dd, 8H, H^o^, ^3^JH^o^/P = 12.6, ^3^JH^o^/H^m^ = 7.4), 7.39 (t, 4H, H^p^, ^3^JH^p^/H^m^ = 7.5), 7.28 (dt, 8H, H^m^, ^3^JH^m^/H^o^ = 7.5, ^3^JH^m^/H^p^ = 7.5, ^4^JH^m^/P = 2.9), 6.98 (d, 2H, H^4^, ^3^JH^4^/H^3^ = 6.5), 6.68 (m, 4H, H^2^H^3^), 6.43 (d, 2H, H^1^, ^3^JH^1^/H^2^ = 8.1), 2.55 (s, 3H, NCH_3_), 2.35 (s, 6H, SCH_3_). ^31^P-{^1^H} NMR (acetone-d_6_, δ/ppm): 6.23 (s, PPh_2_).

### 4.3. Preparation of the Complexes


**Synthesis of 3a**


First the palladium salt must be synthesized “in situ”, adding lithium chloride (21.6 mg, 0.51 mmol) and palladium chloride (45.2 mg, 0.26 mmol) under nitrogen in 15 cm^3^ of deoxygenated methanol for 2 h at 50 °C. After that, the ligand **2a** (84 mg, 0.13 mmol) and sodium acetate (20.9 mg, 0.26 mmol) were added. After stirring for 22 h at 30 °C, a yellow solid was formed, that was centrifugated and dried under vacuum. The solution was evaporated, the residue was dissolved in dichloromethane, filtered under silica and recrystallized from hexane. The yellow solid corresponds to compound **3a**, and the residue obtained from the solution corresponds to compound **4**.

Yield: 39,6 mg, 33%. Anal. calculated: C: 49.8, H: 3.6, N: 3.0, S: 6.8%, found: C: 48.9, H: 3.6, N: 2.9, S: 6.8%; C_39_H_34_Cl_2_N_2_P_2_Pd_2_S_2_ (940.52 g/mol); IR (cm^−1^): υ(P=N) 1298. ^1^H NMR (CDCl_3_, δ/ppm, J/Hz): 7.83 (m, 4H, H^o^), 7.60–7.40 (m, 10H, H^5^/H^6^/H^7^/H^8^/H^p^), 7.34 (d, 2H, H^4^, ^3^JH^4^/H^3^ = 7.7), 7.20 (m, 4H, H^m^), 6.75 (m, 2H, H^2^), 6.61 (m, 2H, H^3^), 6.14 (d, 2H, H^1^, ^3^JH^1^/H^2^ = 8.1), 3.05 (s, 6H, SCH_3_). ^31^P-{^1^H} NMR (CDCl_3_, δ/ppm): 49.40 (s, PPh_2_).


**Product 4**


Yield: 43.2 mg, 36%. Anal. calculated: C: 58.6, H: 4.4, N: 3.5, S: 8.0%, found: C: 58.7, H: 4.4, N: 3.5, S: 8.0%; C_39_H_35_ClN_2_P_2_PdS_2_ (799.66 g/mol); IR (cm^−1^): υ(P=N) 1331, 1297. ^1^H NMR (CDCl_3_, δ/ppm, J/Hz): 8.19 (dd, 4H, Ho, ^3^JH^o^/P = 12.1, ^3^JH^o^/H^m^ = 7.6), 7.79 (m, 4H, H^Ar^), 7.20–7.46 (m, 10H, H^Ar^), 6.79–7.15 (m, 6H, H^Ar^), 6.66 (m, 2H, H^Ar^), 6.37 (d, 1H, H^1^, ^3^JH^1^/H^2^ = 8.1), 2.76 (s, 3H, SCH_3_), 2.54 (s, 3H, SCH_3_). ^31^P-{^1^H} NMR (CDCl_3_, δ/ppm): 51.36 (s, PPh_2_), 41.70 (s, PPh_2_).


**Synthesis of 3b**


The palladium salt was made analogously as above from lithium chloride (18.9 mg, 0.45 mmol) and palladium chloride (39.5 mg, 0.22 mmol). Then, the ligand **2b** (75 mg, 0.11 mmol) and sodium acetate (18.3 mg, 0.22 mmol) were added. After stirring for 22 h at 30 °C, a red solid was formed, which was washed with acetone-dichloromethane and dried under vacuum. Yield: 89,4 mg, 84%. Anal. calculated: C: 50.3, H: 3.8, N: 3.0, S: 6.7%, found: C: 48.8, H: 3.6, N: 2.9, S: 6.3%; C_40_H_36_Cl_2_N_2_P_2_Pd_2_S_2_ (954.55 g/mol); IR (cm^−1^): υ(P=N) 1297. ^1^H NMR (acetone-d_6_, δ/ppm, J/Hz): 7.98 (d, 2H, H^8^, ^3^JH^8^/H^7^ = 8.1), 7.87 (d, 4H, H^o^, ^3^JH^o^/H^m^ = 7.7), 7.56 (t, 2H, H^p^, ^3^JH^p^/H^m^ = 7.5), 7.45 (m, 4H, H^m^, *N* = 7.6), 7.27 (m, 2H, H^6^), 7.12 (d, 2H, H^5^, ^3^JH^5^/H^6^ = 7.9), 7.05 (d, 2H, H^4^, ^3^JH^4^/H^3^ = 7.7), 6.93 (m, 2H, H^7^), 6.87 (m, 2H, H^2^), 6.71 (m, 2H, H^3^), 6.59 (d, 2H, H^1^, ^3^JH^1^/H^2^ = 7.6), 3.53 (m, 4H, PCH_2_), 2.62 (s, 6H, SCH_3_). ^31^P-{^1^H} NMR (acetone-d_6_, δ/ppm): 56.38 (s, PPh_2_).


**Synthesis of 3c**


The palladium salt was made analogously as above from lithium chloride (18.5 mg, 0.44 mmol) and palladium chloride (38.7 mg, 0.22 mmol). Then, the ligand **2c** (75 mg, 0.11 mmol) and sodium acetate (17.9 mg, 0.22 mmol) were added. After stirring for 22 h at 30 °C, an orange solid was formed and was washed with acetone-dichloromethane and dried under vacuum. Yield: 85.7 mg, 81%. Anal. calculated: C: 50.8, H: 4.0, N: 2.9, S: 6.6%, found: C: 49.5, H: 3.8, N: 2.6, S: 6.2%; C_41_H_38_Cl_2_N_2_P_2_Pd_2_S_2_ (968.58 g/mol); IR (cm^−1^): υ(P=N) 1292. ^1^H NMR (acetone-d_6_, δ/ppm, J/Hz): 7.89 (d, 4H, H^o^, ^3^JH^o^/H^m^ = 7.6), 7.84 (d, 4H, H^o^, ^3^JH^o^/H^m^ = 7.6), 7.69–7.41 (m, 24H, H^5^/H^6^/H^7^/H^8^/H^p^/H^4^), 7.15 (m, 8H, H^m^), 7.01 (m, 4H, H^2^), 6.76 (m, 4H, H^3^) 6.56 (d, 2H, H^1^, ^3^JH^1^/H^2^ = 8.3), 6.52 (d, 2H, H^1^, ^3^JH^1^/H^2^ = 8.3), 3.34 (m, 8H, PC*H_2_*), 2.65 (s, 6H, SCH_3_), 2.63 (s, 6H, SCH_3_). ^31^P-{^1^H} NMR (acetone-d_6_, δ/ppm): 57.99 (s, PPh_2_), 57.25 (s, PPh_2_).


**Synthesis of 3d**


The palladium salt was made analogously as above from lithium chloride (20.2 mg, 0,48 mmol) and palladium chloride (42.3 mg, 0.24 mmol). Then, the ligand **2d** (80 mg, 0.12 mmol) and sodium acetate (19.6 mg, 0.24 mmol) were added. After stirring for 22 h at 30 °C, a brown solid was formed that was washed with acetone-dichloromethane and dried under vacuum. Yield: 92.9 mg, 82%. Anal. calculated: 50.4, H: 3.6, N: 2.9, S: 6.7%, found: C: 49.8, H: 3.8, N: 3.0, S: 6.3%; C_40_H_34_Cl_2_N_2_P_2_Pd_2_S_2_ (952.53 g/mol); IR (cm^−1^): υ(P=N) 1291. ^1^H NMR (acetone-d_6_, δ/ppm, J/Hz): 8.00 (m, 2H, H^8^), 7.72–7.52 (m, 18H, H^4^/H^5^/H^6^/H^7^/H^o^/H^m^/H^p^), 6.89 (m, 2H, H^2^), 6.73 (m, 4H, H^1^/H^3^), 4.32 (m, 1H, PCCH_2_), 3.86 (m, 1H, PCCH_2_), 2.73 (s, 6H, SCH_3_). ^31^P-{^1^H} NMR (acetone-d_6_, δ/ppm): 52.72 (s, PPh_2_).


**Synthesis of 3e**


The palladium salt was made analogously as above from lithium chloride (8.8 mg, 0.21 mmol) and palladium chloride (18.4 mg, 0.10 mmol). Then, the ligand **2e** (35 mg, 0.05 mmol) and sodium acetate (8.5 mg, 0.10 mmol) were added. After stirring for 22 h at 30 °C, a green solid was formed that was washed with acetone-dichloromethane and dried under vacuum. Yield: 32.8 mg, 66%. Anal. calculated: C: 49.0, H: 3.7, N: 4.4, S: 6.7%, found: C: 47.2, H: 3.3, N: 4.6, S: 6.5%; C_39_H_35_Cl_2_N_3_P_2_Pd_2_S_2_ (955.54 g/mol); IR (cm^−1^): υ(P=N) 1299. ^1^H NMR (acetone-d_6_, δ/ppm, J/Hz): 8.19 (d, 2H, H^8^, ^3^JH^8^/H^7^ = 8.0), 8.09 (d, 4H, H^o^, ^3^JH^o^/H^m^ = 8.6), 7.77 (t, 2H, H^p^, ^3^JH^p^/H^m^ = 8.5), 7.60 (m, 4H, H^m^, *N* = 8.6), 7.45 (m, 2H, H^6^), 7.37 (m, 2H, H^5^), 7.29 (d, 2H, H^4^, ^3^JH^4^/H^3^ = 7.3), 7.16 (m, 2H, H^7^), 7.07 (m, 2H, H^2^), 6.98 (m, 2H, H^3^), 6.77 (d, 2H, H^1^, ^3^JH^1^/H^2^ = 7.3), 2.71 (s, 6H, SCH_3_), 2.63 (s, 3H, NCH_3_). ^31^P-{^1^H} NMR (acetone-d_6_, δ/ppm): 48.41 (s, PPh_2_).


**Product 5**


Compound **3a** (94.0 mg, 0.10 mmol), dppe (19.9 mg, 0.05 mmol) and NH_4_PF_6_ (16.3 mg, 0.1 mmol) were added in acetone (15 cm^3^). The resulting mixture was stirred at room temperature for 18 h, and an untreatable solid was formed which was filtered off. The solid was dissolved in chloroform, which produced single crystals of compound **5**.

### 4.4. Crystal Structure Analysis and Details on Data Collection and Refinement

CCDC 2113724(**3c**) and 2114034(**5**). See Appendix A.

## Data Availability

Not applicable.

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
