# Peer review of "Cyclometallated Palladium(II) Complexes: An Approach to the First Dinuclear Bis(iminophosphorane)phosphane-[C,N,S] Metallacycle"

_molecules, 2022, doi:10.3390/molecules27207043_

Round 1
Reviewer 1 Report (New Reviewer)
The paper is interesting. However, some minor points should be corrected. A correction of the English form is required in the text. Here I give some suggestions.
At line 130
The compound crystallizes in orthorhombic space group Pnma ….
The crystals consist of discrete molecules separated by van der Waals distances, bearing two palladacycle moieties symmetry related by a 2-fold axis passing through the C(21) carbon atom of the iminophosphorane ligand. The crystallographic independent palladium atom is bonded to four different atoms in a slightly distorted square‒planar disposition: a P=N nitrogen atom, an aryl carbon atom, a sulfur from the thiomethyl aniline, and finally by a chloride ligand. The metal coordination planes [C,N,S,Cl] form a dihedral angle of 22.24˚.
DELETE : “The structure is crystallographically centrosymmetric with the inversion center situated at the C(21) carbon between the two phosphorus atoms.” THE STRUCTURE IS NOT CENTROSYMMETRIC, BUT AS REPORTED ABOVE HAS A 2-FOLD SYMMETRY
IF CENTROSYMMETRIC ANGLE BETWEEN COORDINATION PLANES SHOULD BE 0° !
At line 158
…in acetone under stirring condition for 18 h at room temperature gave a sticky solid, for which the spectroscopic data indicated to be a complex mixture of products, which were not further pursued. However, the chloroform solution of this solid gave suitable crystals for X-ray structural analysis that proved to be not the expected product with the diphosphane bonded to two palladium atoms, but rather a different and new coordination compound, labelled 5. (see 164 Scheme 2).
At line 172
Compound 5 crystallizes in monoclinic space group P21/n (NOTE: number “1” should be SUBSCRIPT) with two chloroform solvent molecules;…
line 176
The sum of coordination bond angles at palladium is 359.67˚.
At line 218
Attempt to prepare complex 5 by direct treatment of 2a and an appropriate palladium salt was unsuccessful to reproduce the result here reported. Further…
Better to say:
Figure 1. Ortep drawing of complex 3c with thermal ellipsoid drawn at 50% probability level.
Figure 2. Ortep drawing of complex 5 with thermal ellipsoid drawn at 50% probability level.
Author Response
All the corrections have been embedded in the manuscript template sent by the editor. This manuscript file has been uploaded in the Molecules web page.
Reviewer 2 Report (New Reviewer)
I have several comments of different levels to the text -
1. Different number of decimal places, why different accuracy?
2. Why in the article first there are conclusions, and after there is a description of the synthesis?
3. Where is the list of chemicals?
4. The quality of the pictures is extremely low, can you improve the quality?
5. Figure captions do not describe what is depicted on them.
6. Where is description of NMR, IR and X-ray experiments and experimental data?
7. Where is CCDC numbers? Does this structures already deposited, or not? I want to have opportunity to check your cif files.
8. Why was this ratio chosen for the reagents? For example 1:1 for LiCl and PdCl2.
Author Response
- Different number of decimal places, why different accuracy? Microanalytical data needs only one decimal figure as the absolute error is ca. 0.3%. However, for NMR data the resonances may be given with two decimal figures. This accounts for the apparent difference in accuracy which stems from the differing precision of the techniques.
2. Why in the article first there are conclusions, and after there is a description of the synthesis?
This follows the Molecules template for manuscripts.
3. Where is the list of chemicals?
At the beginning of the Experimental there is a paragraph with the chemicals employed in the preparations.
4. The quality of the pictures is extremely low, can you improve the quality?
The pictures have been changed to give a better quality. These are now in the re-submitted manuscript file through the Molecules web page.
5. Figure captions do not describe what is depicted on them.
All the captions tell what the corresponding figure is about. The ensuing explanation is then given in the text. We believe that otherwise it would make it rather redundant.
6. Where is description of NMR, IR and X-ray experiments and experimental data?
Under each preparation in the Experimental a detailed account of the yield, microanalytical data and spectroscopic results is given.
7. Where is CCDC numbers? Does this structures already deposited, or not? I want to have opportunity to check your cif files.
The structures have been adequately deposited in the CCDC and the numbers are now specifically included in the manuscript in part 4.4. The check CIF files have been previously corrected and reviewed in previous revisions of this manuscript. The CIF file for compound 3c has no A and B alerts. The CIF file for compound 5 has only one B alert. We have gone through the CCDC data base, and in Alert B we have a included a comment on this point. The response was: Density around at 1.76Ang can be attributed to residual water molecule, but could not be assigned.
8. Why was this ratio chosen for the reagents? For example 1:1 for LiCl and PdCl2.
This has been corrected in the amended manuscript.
Round 2
Reviewer 2 Report (New Reviewer)
last questions -
about question 1.
Different number of decimal places, why different accuracy? - There are still varying accuracy, for example - Pd(1)‐N(2) 2.079(2) Pd(1)‐S(1) 2.2655(8). I'm talking only about X-ray data here.
Correct form must looks like -Pd(1)‐N(2) 2.0790(20) Pd(1)‐S(1) 2.2655(8). It's just rules of presentation.
about question 7.
You can improve your structure and exclude alert B by yousing SQUEEZE function. Just cut incorrect electron density from your structure.
Author Response
about question 1.
Different number of decimal places, why different accuracy? - There are still varying accuracy, for example - Pd(1)‐N(2) 2.079(2) Pd(1)‐S(1) 2.2655(8). I'm talking only about X-ray data here.
Correct form must looks like -Pd(1)‐N(2) 2.0790(20) Pd(1)‐S(1) 2.2655(8). It's just rules of presentation.
This has been amended and is now included in the new version of the manuscrip.
about question 7.
You can improve your structure and exclude alert B by yousing SQUEEZE function. Just cut incorrect electron density from your structure.
The structure for compound 5 has been corrected and the check CIF shows there are no B alerts. We include the two check CIF files in the submission
This manuscript is a resubmission of an earlier submission. The following is a list of the peer review reports and author responses from that submission.
Round 1
Reviewer 1 Report
The article “Cyclometallated palladium(II) complexes: an approach to the first dinuclear bis(iminophosphorane)phosphane-[C,N,S] metallacycle” by J. M. Vila and co-authors reports synthesis of a set of bis(iminophosphorane)phosphanes with different linker between donor sites and a family of binuclear bridging Pd(II) complexes based on them. A reactivity of one Pd(II) complex toward diphosphine was demonstrated without details.
The composition and structure of bis(iminophosphorane)phosphanes as well as of Pd(II) complexes in solution were established based on CHN analysis, polynuclear NMR spectroscopy (1H and 31P) and FTIR-spectroscopy. Solid state structure of two Pd(II) complexes were determined based on single crystal X-ray analysis.
The article reports routine studies that are interesting as some expanding of family of well known iminophosphorane-phosphanes and their complexes with transition metals, and cannot be recommend it for publication in Molecules in the current form. As is the article could be recommended for publication in Zeitschrift für anorganische und allgemeine Chemie or related journal.
In the first of all, this decision is based on the fact that the completely analogues mononuclear Pd(II), Pt(II) and Au(III) complexes are already reported (see, for example DOI: 10.1016/j.jorganchem.2013.09.038; 10.1016/j.jorganchem.2016.04.022). The second main reason is that no any chemical of physicochemical properties (catalysis, photoluminescence, biological activity, etc.) of the Pd(II) complexes are demonstrated. The only one example of only one complex reactivity toward to only one diphosphine with unclear result is not enough.
On the other hand, to make the material suitable for publication in Molecules in the special issue “Organometallic Complexes: Fundamentals and Applications” the following additional studies could be recommended. It looks very interesting to complete the investigation of reported Pd(II) complexes reactivity toward to eponymous diphosphines with the target to substitute chlorine ligands and to obtain “closed” binuclear Pd(II) complexes (for all complexes 3a-3e). Using of the same diphosphine with the same distance between donor sites could result in obtaining of Pd(II) complexes with Pd−Pd metallophilic intramolecular interaction that could be really interesting and actual chemistry.
Besides that, the following points should be answer:
(a) References DOI: 10.1039/C6DT04592H; 10.1021/acs.organomet.8b00867; 10.1016/j.jorganchem.2016.04.022; 10.1016/j.ica.2009.04.024 could be recommended to include.
(b) Scheme 2 should be draw in better resolution.
(c) All spectra (FTIR, 1H and 31P NMR) for all compounds should be placed in Supplementary Information (SI).
(d) Numeration of atoms for protons assignment should be given for all compounds, and in the same way with Experimental and Discussion. Numerations for compound 1 in Scheme 2 and in Experimental, for example, are different, and there is no proton number 5 (page 4, line 113) on Scheme 2 as well. It make sense to place chemical structures with complete numeration in SI.
(e) Atom’s labels in Figure 1 and 2 are too small.
(f) Experimental section:
− “MeCOMe-d6” looks strange. Please, change it for “acetone-d6” or “(CD3)2CO” according with traditional manner.
− Paragraph 4.2 looks as “Preparation of bis(iminophosphorane)phosphane” not the complexes. Paragraph 4.3 named “Preparation of Pd(II) complexes” should be organized for description of complexes synthesis.
− X-ray structure determinations description should be included.
− IUPAC names for organic compounds should be included.
− Synthetic procedures for compounds 2 and for compounds 3 look the same. It is enough to detail them once and then refer to it.
− Description of reaction of 3a with dppe should be include.
(g) List of References is wrong formatted.

Author Response
Referee 1
The article “Cyclometallated palladium(II) complexes: an approach to the first dinuclear bis(iminophosphorane)phosphane-[C,N,S] metallacycle” by J. M. Vila and co-authors reports synthesis of a set of bis(iminophosphorane)phosphanes with different linker between donor sites and a family of binuclear bridging Pd(II) complexes based on them. A reactivity of one Pd(II) complex toward diphosphine was demonstrated without details. The composition and structure of bis(iminophosphorane)phosphanes as well as of Pd(II) complexes in solution were established based on CHN analysis, polynuclear NMR spectroscopy (1H and 31P) and FTIR-spectroscopy. Solid state structure of two Pd(II) complexes were determined based on single crystal X-ray analysis. The article reports routine studies that are interesting as some expanding of family of well known iminophosphorane-phosphanes and their complexes with transition metals, and cannot be recommend it for publication in Molecules in the current form. As is the article could be recommended for publication in Zeitschrift für anorganische und allgemeine Chemie or related journal. In the first of all, this decision is based on the fact that the completely analogues mononuclear Pd(II), Pt(II) and Au(III) complexes are already reported (see, for example DOI: 10.1016/j.jorganchem.2013.09.038; 10.1016/j.jorganchem.2016.04.022). The second main reason is that no any chemical of physicochemical properties (catalysis, photoluminescence, biological activity, etc.) of the Pd(II) complexes are demonstrated. The only one example of only one complex reactivity toward to only one diphosphine with unclear result is not enough. On the other hand, to make the material suitable for publication in Molecules in the special issue “Organometallic Complexes: Fundamentals and Applications” the following additional studies could be recommended. It looks very interesting to complete the investigation of reported Pd(II) complexes reactivity toward to eponymous diphosphines with the target to substitute chlorine ligands and to obtain “closed” binuclear Pd(II) complexes (for all complexes 3a-3e). Using of the same diphosphine with the same distance between donor sites could result in obtaining of Pd(II) complexes with Pd−Pd metallophilic intramolecular interaction that could be really interesting and actual chemistry.
We are aware that similar iminophosphorane compounds have been described. One of the DOI addresses the Reviewer mentions, 10.1016/j.jorganchem.2013.09.038, is precisely our reference [31] already in the submitted version. As is explained in the Introduction several reasons are given to show the difference between the compounds reported herein, and the already known iminopohsphorane species. So, one may consider the present compounds may be analogous and/or similar but surely they are different and new.
In relation to the special issue “Organometallic Complexes: Fundamentals and Applications”, a manuscript may be related to Fundamentals, Applications or both. Our present research deals with the Fundamentals, so we believe that the manuscript fits the title description perfectly.
As for the idea of completing the investigation and as we express in the manuscript, of course much more needs to be done; however, each stage of the investigation must be limited. So, when a certain amount of experimental data and results have been obtained and checked, publication is mandatory. One could perform many other additional experiments and comply with the corresponding suggestions, but it would be a never-ending story.
Besides that, the following points should be answer:
(a) References DOI: 10.1039/C6DT04592H; 10.1021/acs.organomet.8b00867; 10.1016/j.jorganchem.2016.04.022; 10.1016/j.ica.2009.04.024 could be recommended to include.
These references have been included as numbers 34, 35 and 36.
(b) Scheme 2 should be draw in better resolution.
A new Scheme has been included, both in the manuscript and in the SI.
(c) All spectra (FTIR, 1H and 31P NMR) for all compounds should be placed in Supplementary Information (SI).
Regarding the “live” IR and NMR spectra, the Experimental Section contains detailed information of the spectra and there is really no need to include the spectra themselves in the text nor in the SI. This is very journal space consuming; also, and with all due respect, it makes you feel that the Reviewer doesn´t trust us. Usually “live” spectra are imbedded for kinetic studies and the like.
(d) Numeration of atoms for protons assignment should be given for all compounds, and in the same way with Experimental and Discussion. Numerations for compound 1 in Scheme 2 and in Experimental, for example, are different, and there is no proton number 5 (page 4, line 113) on Scheme 2 as well. It make sense to place chemical structures with complete numeration in SI.
A new Scheme has been included in the manuscript and in the SI. The latter shows the ortho, meta and para protons.
(e) Atom’s labels in Figure 1 and 2 are too small.
These have been amended.
(f) Experimental section:
− “MeCOMe-d6” looks strange. Please, change it for “acetone-d6” or “(CD3)2CO” according with traditional manner.
This has been changed to acetone-d6.
− Paragraph 4.2 looks as “Preparation of bis(iminophosphorane)phosphane” not the complexes. Paragraph 4.3 named “Preparation of Pd(II) complexes” should be organized for description of complexes synthesis.
These changes have been made with the new titles for the paragraphs.
− X-ray structure determinations description should be included.
This is in the SI.
− IUPAC names for organic compounds should be included.
The compounds herein are inorganic or organometallic. 2-methylthioaniline now bears the IUPAC name 2-(methylsulfanyl)aniline.
− Synthetic procedures for compounds 2 and for compounds 3 look the same. It is enough to detail them once and then refer to it.
Indeed they do, the reason for depicting this way is that the quantities of the products and the colors of the resulting solids are different.
− Description of reaction of 3a with dppe should be include.
This has been included.
(g) List of References is wrong formatted.
This has been corrected.

Reviewer 2 Report
This manuscript is the review described about cyclometallated dipalladium complexes with bisimonophosphanes bridging a diphosphine moiety. The system of reaction and obtained structures are much attractive and obtained products are also characterized definitely. The manuscript should be published but before that, it would be grateful if you answer and/or reflect the feedbacks in the manuscript if necessary.
(1) About the phenomenon that the reaction with 2a and palladium sources is different from that with 2b-2e, the authors supposed in manuscript, “the differing behavior of ligand 2a as opposed to 3b-3e could be due to certain steric hindrance imposed by the shorter P-P carbon chain.” However, both 2d and 2e are also diphosphines bridged by “only one atom” as well as 2a, so the similar reactions may occur on utilization of these iminophosphanes, I think. Is it certain it stems from only shorter chain?
(2) About the assignment of 31P NMR in complex 4, the authors said, “For complex 4 there were two singlet resonances at 41.70 and 51.36 ppm in the 31P NMR spectrum, consequent of two non-equivalent phosphorus nuclei.” However, chemical shift of phosphorus in free ligand 2a is 22.37. This showed that the signal of non-coordinated phosphorus atom is shifted to lower field by at least about 15 ppm. Though environment around non-coordinated phosphorus atom may be actually varied, the shift is too large even if so, I think. I doubt a little that complex 4 becomes this mono palladium structure also in solution.
In addition, I will tell the authors the points leading to misreading.
In line 92, “solvent extraction” may be “solvent exclusion”. (“evaporation” is descripted in experimental section)
In line 112, which is H5 in “H5 resonance”? In experimental section, there are also the notations, H7, H8 and so on. I could not recognize which position of hydrogen each number show in the compounds. About the position of hydrogen, another notation should be examined.
Author Response
Referee 2
This manuscript is the review described about cyclometallated dipalladium complexes with bisimonophosphanes bridging a diphosphine moiety. The system of reaction and obtained structures are much attractive and obtained products are also characterized definitely. The manuscript should be published but before that, it would be grateful if you answer and/or reflect the feedbacks in the manuscript if necessary.
(1) About the phenomenon that the reaction with 2a and palladium sources is different from that with 2b-2e, the authors supposed in manuscript, “the differing behavior of ligand 2a as opposed to 3b-3e could be due to certain steric hindrance imposed by the shorter P-P carbon chain.” However, both 2d and 2e are also diphosphines bridged by “only one atom” as well as 2a, so the similar reactions may occur on utilization of these iminophosphanes, I think. Is it certain it stems from only shorter chain?
This explanation was meant mainly for compounds 3a-3c having differing saturated carbon chain lengths. For 3d the PCP carbon is an sp2 carbon, hence with a greater PCP bond angle of 120°, cf. 109,5° 2d, thus lowering steric hindrance. As for 3e, we suggest this could be attributed to a greater flexibility of the PNP fragment leading only to the dinuclear complexes.
(2) About the assignment of 31P NMR in complex 4, the authors said, “For complex 4 there were two singlet resonances at 41.70 and 51.36 ppm in the 31P NMR spectrum, consequent of two non-equivalent phosphorus nuclei.” However, chemical shift of phosphorus in free ligand 2a is 22.37. This showed that the signal of non-coordinated phosphorus atom is shifted to lower field by at least about 15 ppm. Though environment around non-coordinated phosphorus atom may be actually varied, the shift is too large even if so, I think. I doubt a little that complex 4 becomes this mono palladium structure also in solution.
We agree with the Reviewer, and at this moment we need to further research this issue as was already mentioned in the text, in spite that the microanalytical data are in accordance with the proposed formula. Nevertheless, a short paragraph commenting this has been added to the manuscript.
In addition, I will tell the authors the points leading to misreading.
In line 92, “solvent extraction” may be “solvent exclusion”. (“evaporation” is descripted in experimental section)
This has now been amended in the text, and changed to solvent removal.
In line 112, which is H5 in “H5 resonance”? In experimental section, there are also the notations, H7, H8 and so on. I could not recognize which position of hydrogen each number show in the compounds. About the position of hydrogen, another notation should be examined.
The Scheme has been amended accordingly with the missing notations and included manuscript.

Reviewer 3 Report
The manuscript-1799705 “Cyclometallated palladium(II) complexes: an approach to the first dinuclear bis(iminophosphorane)phosphane-[C,N,S] metallacycle” by José M. Vila*, MARCOS RUA-SUEIRO, PAULA MUNIN-CRUZ, ADOLFO FERNANDEZ , Juan M. Ortigueira, M. Teresa Pereira concerns the synthesis and investigation unprecedented bis(iminophosphorane) phosphane-[C,N,S] palladacycles. This pioneering study is narrowly specific and of interest only to scientists involved in the coordination and organometallic chemistry of palladium and their practical applications have not been investigated. In addition, the majority of compounds have not been fully characterized. Of the six complexes, only two were structurally characterized, and X-ray phase analysis confirming the individuality of the compounds was not performed for any of the complexes.
Unfortunately, I cannot recommend this work for publication in MOLECULES, which is a general journal.
It is necessary to add a comment on alert B to the cif-file for connection 3c.
The authors claim that they have obtained a new family of complexes, although only one member has been structurally characterized. At the same time, there are no "live" IR and NMR spectra in the Supporting information, which greatly complicates the perception of the paper text and its understanding.
Explain what causes a slight co-directional decrease in the content of carbon, nitrogen and sulfur in compound 1.
Explain what causes a slight co-directional decrease in the content of all analyzed in compound 2b.
Ligands 2a - 2e were not subjected to a differentiating procedure, which affected their purity and the results of elemental analysis.
X-ray diffraction analysis was performed only for two compounds; for the rest, even crystallinity and phase purity were not confirmed by X-ray phase analysis.
There is a significant underestimation of the content of C and S, with almost ideal values of H and N for compound 2d.The elemental analysis for 2e is also imperfect.
It is not clear how the authors calculated the yield for compounds 3a and 4, since they were obtained from a single batch of starting reagents.
Author Response
Referee 3
The manuscript “Cyclometallated palladium(II) complexes: an approach to the first dinuclear bis(iminophosphorane)phosphane-[C,N,S] metallacycle” by José M. Vila*, MARCOS RUA-SUEIRO, PAULA MUNIN-CRUZ, ADOLFO FERNANDEZ , Juan M. Ortigueira, M. Teresa Pereira concerns the synthesis and investigation unprecedented bis(iminophosphorane) phosphane-[C,N,S] palladacycles. This pioneering study is narrowly specific and of interest only to scientists involved in the coordination and organometallic chemistry of palladium and their practical applications have not been investigated. In addition, the majority of compounds have not been fully characterized. Of the six complexes, only two were structurally characterized, and X-ray phase analysis confirming the individuality of the compounds was not performed for any of the complexes.
Unfortunately, I cannot recommend this work for publication in MOLECULES, which is a general journal.
It is necessary to add a comment on alert B to the cif-file for connection 3c.
We have gone through the CCDC data base, and in Alert B we have a included commented on this point. The response was: Density around at 1.76Ang can be attributed to residual water molecule, but could not be assigned.
The authors claim that they have obtained a new family of complexes, although only one member has been structurally characterized. At the same time, there are no "live" IR and NMR spectra in the Supporting information, which greatly complicates the perception of the paper text and its understanding.
One can prepare a new family of complexes of any type without having to do the crystallographic analysis for all of them. In the present case all the complexes 3a to 3e, plus 4 and 5, are all new and X-ray analysis has been fulfilled for two of them. Regarding the “live” IR and NMR spectra, the Experimental Section contains detailed information of the spectra and there is really no need to include the spectra themselves in the text nor in the SI. This is very journal space consuming; also it makes you feel that the Reviewer doesn´t trust us. Usually “live” spectra are imbedded for kinetic studies and the like.
Explain what causes a slight co-directional decrease in the content of carbon, nitrogen and sulfur in compound 1.
This has more to do with the experimental technique than with the compound itself. The awkward situation would be to have ups and downs in the results. Nevertheless, we must remember that the experimental error for microanalysis is 0.3% absolute, so we are well within the allowed limits.
Explain what causes a slight co-directional decrease in the content of all analyzed in compound 2b.
Same reply as above.
Ligands 2a - 2e were not subjected to a differentiating procedure, which affected their purity and the results of elemental analysis.
All the ligands were prepared by a similar procedure, which gave satisfactory results. We found no justification for changing procedures that worked well.
X-ray diffraction analysis was performed only for two compounds; for the rest, even crystallinity and phase purity were not confirmed by X-ray phase analysis.
This has been commented above. Recrystallizing compounds is a trial and error issue. One cannot make a compound, even if it is a crystalline powder, to give single-crystals.
There is a significant underestimation of the content of C and S, with almost ideal values of H and N for compound 2d. The elemental analysis for 2e is also imperfect.
This has been commented above.
It is not clear how the authors calculated the yield for compounds 3a and 4, since they were obtained from a single batch of starting reagents.
In the experimental part it is detailed that two compounds are obtained. Once separated and weighed, they are referenced to the initial weight of the starting compound and then the yield is calculated.

Round 2
Reviewer 1 Report
The article “Cyclometallated palladium(II) complexes: an approach to the first dinuclear bis(iminophosphorane)phosphane-[C,N,S] metallacycle” by J. M. Vila and co-authors reports synthesis of a set of bis(iminophosphorane)phosphanes with different linker between donor sites and a family of binuclear bridging Pd(II) complexes based on them. A reactivity of one Pd(II) complex toward an one diphosphine was demonstrated.
The authors have improved the text of the article, and the reviewer agrees rather than disagree with arguments made about the nature of the research described. In the present case, however, the term “new compounds” takes priority over the term “novel compounds”.
The following points should still be answer:
(a) Page 2, Line 35. “More often than not they are synthesized by the Staudinger reaction.” Change this sentence because it is not clear.
(b) With all my respect to authors, I keep insisting that the spectra (FTIR, 1H and 31P NMR) be included in Supplementary Information (SI). It is normal practice for synthetic work, and size of SI is not limited.
(c) IUPAC names for compounds 2a-2e also should be included in Experimental section.
(d) The spectral data for compound 2a invites some questions. The phosphorous chemical shift in NMR spectrum is drastically different from that for 2b-2e and very close to phosphorous oxide (P=O group). How can one make sure that 2a contains a P=N group like for the other compounds 2?
(e) The phosphorous chemical shifts in NMR spectrum of complex 4 look strange also. If it is mononuclear non-symmetrical complex with one free P=N donor site, why the both phosphorous resonances are low-shifted in compare with 2a, not just one of the two?
(f) Compound 2e is sterically closed to 2a and even less flexible. Why is there no problem in obtaining the binuclear complex 3e?
(g) As a suggestion. Perhaps a light gray color for the carbon atoms in Figures 1 and 2 would be better than blue.
(h) List of References is still wrong formatted.
Reviewer 3 Report
The authors were disrespectful to the reviewers by providing a PDF version of the Word file in correction mode, instead of simply highlighting the changed elements in the revised version of the manuscript.
The authors have ignored the comments regarding inaccuracies in the manuscript English, which were noted directly in the attached manuscript file. As well, some remarks in the experimental part. For example, a comment about the absence of the of hydrochloric acid concentration in synthesis of the compound 1.
To the great regret of the reviewer, the authors unconvincingly have responded to the comments concerning the study, but the remarks related to the English inaccuracies were simply ignored by them.
Responding to the remark about the phase purity of the studied complexes in the solid state, at least for structurally characterized complexes, the authors have demonstrated gross ignorance that powder X-ray diffraction is the main method for establishing phase purity for substances in the solid state. Only if the experimental diffraction pattern coincides with the theoretical one simulated from the data for a single crystal, it can be argued that the sample is monophasic. This is important in the presence of precipitation of a mixture of modifications of the same compound, or a possible realization of any type of isomerism.
Based on the foregoing, the reviewer did not change his mind.
